# Comparison between passive knee kinematics during surgery and active knee kinematics during walking: A preliminary study

**Xavier Gasparutto**[1]*, **Alice Bonnefoy-Mazure**[1], **Michael Attias**[2], **Raphaël Dumas**[3], **Stéphane Armand**[1], **Hermès Miozzari**[4]

**1** Kinesiology Laboratory, Geneva University Hospitals and University of Geneva, Geneva, Switzerland, **2** School of Health Sciences, University of Applied Sciences and Arts Western Switzerland (HES-SO), Geneva, Switzerland, **3** Claude Bernard University Lyon 1 and Gustave Eiffel University, LBMC UMR_T9406, University of Lyon and Laboratory of Biomechanics and Impact Mechanics, Lyon, France, **4** Department of Orthopaedic Surgery and Trauma Care, Geneva University Hospitals, Geneva, Switzerland

\* xavier.gasparutto@hcuge.ch

**Data Availability Statement:** All data are shared on the online repository Yareta (https://doi.org/10. 26037/yareta:33ylesvms5heboezppb7nkcpau).

## Abstract

Recovery of function is among a patient's main expectations when undergoing total knee arthroplasty (TKA). However, normal gait knee function is not always completely restored, which can affect patient satisfaction and quality of life. Computer-assisted surgery (CAS) allows surgeons to evaluate passive knee kinematics intra-operatively. Understanding associations between knee kinematics measured during surgery and during daily activities, such as walking, could help define criteria for success based on knee function and not only on the correct alignment of the implant or the leg. This preliminary study compared passive knee kinematics measured during surgery with active kinematics measured during walking. Eight patients underwent a treadmill gait analysis using the KneeKG™ system both before surgery and three months afterwards. Knee kinematics were measured during CAS both before and after TKA implantation. The anatomical axes of the KneeKG™ and CAS systems were homogenised using a two-level, multi-body kinematics optimisation with a kinematic chain based on the calibration measured during CAS. A Bland–Altman analysis was performed before and after TKA for adduction–abduction angle, internal–external rotation, and anterior–posterior displacement over the whole gait cycle, at the single stance phase and at the swing phase. Homogenising the anatomical axes between CAS and treadmill gait led to limited median bias and limits of agreement (post-surgery -0.6 ± 3.6 deg, -2.7 ± 3.6 deg, and -0.2 ± 2.4 mm for adduction–abduction, internal–external rotation and anterior–posterior displacement, respectively). At the individual level, correlations between the two systems were mostly weak ($R^2 < 0.3$) over the whole gait cycle, indicating low kinematic consistency between the two measurements. However, correlations were better at the phase level, especially the swing phase. The multiple sources of differences did not enable us to conclude whether they came from anatomical and biomechanical differences or from measurement system errors.

Code are shared on GitLab (https://gitlab.unige.ch/KLab).

**Funding:** This work was supported by Geneva University Hospitals' Department of Orthopaedic Surgery and Trauma Care. The funding source played no role in the study's design.

**Competing interests:** I have read the journal's policy and the authors of this manuscript have the following competing interests: Hermes Miozzari is an associate editor for EFORT Open Reviews and a board member of Swiss Orthopaedics. Stéphane Armand is a member of the editorial board of EFORT Open Reviews. There are no competing interests associated with this research.

## Introduction

The primary expectation for patients undergoing total knee arthroplasty (TKA) is pain relief. The four next most frequently expressed expectations are all linked to the recovery of function: mobility, walking, physical activities and daily activities [1]. However, TKA does not always fully restore knee function [2], which patients may experience as a failure, with direct effects on their satisfaction and quality of life [3]. In this context, several tools and technologies have been developed to support surgeons in their practice and to assess the knee joint's functional behaviour during surgery (passive movements) and functional tasks such as walking (active movements). Computer-assisted surgery (CAS) was developed to help plan and execute surgical interventions [4]; its goal is to improve the accuracy and precision of component positioning and leg alignment [4]. More recent CAS systems can also evaluate passive knee kinematics and stability and provide real-time feedback on multiple parameters, such as implant positioning and overall alignment, passive knee range of motion (RoM), ligament balancing and knee laxity during surgery. Fluoroscopy and optoelectronic motion-capture systems are the current standards for assessing active knee movements before and after TKA. Contrary to optoelectronic methods, however, fluoroscopy is not affected by soft-tissue artefacts (STAs) and is thus the gold standard for evaluating active knee function. However, fluoroscopy is highly invasive due to the large dose of radiation received during measurement and the need to perform a CT scan of the lower limb. This method is mainly used for research purposes and, in clinical settings, optoelectronic systems such as the KneeKG™ [5] are the standard means of measuring knee function. The KneeKG™ system was developed to provide patients with a knee kinesiography examination during walking [6], i.e. a measurement of knee kinematics in mobility-impaired patients. This system could be considered the silver standard for knee assessment as it provides surgeons with a rapid understanding of a patient's knee kinematics and gait before and after TKA [6]. It is an approved medical device in Europe (CE mark class IIa) and the United States of America (FDA 510(k) clearance). Understanding whether there is an association between passive knee kinematics measured during surgery (using CAS) and knee kinematics measured during daily activities such as walking (using KneeKG™) could help surgeons get closer to reproducing normal knee function during daily activities rather than focussing solely on component alignment. Grassi *et al.* recently showed that there were no significant differences between active and passive intra-operative knee kinematics as measured using a CAS system during knee flexion–extension before and after TKA [7]. Belvedere *et al.* showed that passive knee kinematics measured using CAS before wound closure, with the definitive TKA component in place, was predictive of post-surgical kinematics measured using monoplane fluoroscopy during multiple weight bearing activities (stair ascent, chair standing and sitting) and during flexion–extension against gravity, six months after surgery [8, 9].

However, to the best of our knowledge, no studies have analysed the link between CAS and KneeKG™ measurements before and after TKA. Since this method is the standard in clinical practice, it seems relevant to explore associations between intra-operative kinematics and gait assessment pre- and post-TKA.

This preliminary study investigated potential associations between passive knee kinematics, measured using CAS before and after TKA, and active knee kinematics measured during treadmill gait analysis using the KneeKG™ system, before and three months after surgery. Data were compared in terms of corresponding (1) anatomical axes, (2) patterns and (3) variability.

## Materials and methods

### Participant selection and characteristics

All patients scheduled for a primary unilateral TKA for symptomatic end-stage osteoarthritis (OA, i.e. grade III to IV on the Kellgren–Lawrence classification) at our tertiary care centre between 2019 and 2020 were asked to participate in this preliminary study. Patient exclusion criteria were previous lower-limb arthroplasty, a history of lower-limb or lower-back surgery, neurological or orthopaedic disorders that could affect gait or balance, and the use of crutches or any walking aid. The local ethics committee approved the study (n. CCER 2018–00819). Written informed consent was obtained from all participants.

All the TKAs were performed by a senior surgeon (HHM), using a standard medial parapatellar approach and a routine measured resection technique, with either a posterior stabilised or a medial-pivot TKA design. CAS was only used to record passive motion. Four patients had their patella resurfaced. All components were fixed using bone cement (polymethylmethacrylate). The location of the patient's knee OA (the medial tibiofemoral, lateral tibiofemoral and patellofemoral compartments) was assessed before TKA using weight-bearing antero–posterior and lateral X-rays as well as a skyline view of the patella. In addition, lower-limb alignment was quantified using the hip–knee–ankle angle from standing long-leg X-rays. These values were measured by an experienced orthopaedic surgeon (HHM).

### Computer-assisted surgery measurements

During surgery, intra-cortical pins with reflective markers were implanted in the femur and tibia after arthrotomy, and a standard medial release (as part of the approach) left the central pivot untouched. An anatomical calibration of the lower limb was then performed. Measures recorded before the definitive fixation of the TKA (OA conditions) and after the definitive fixation of the TKA component (before closing the arthrotomy) included passive flexion–extension movements, with and without adduction–abduction stress, as well as internal–external rotation and anterior–posterior translation at various degrees of flexion of the lower limb. The present study only considered passive flexion–extension movements without adduction–abduction stress. The CAS system routinely used was the Knee 3 (Brainlab, Munich, Germany), with a customised 'record' button employed for specific research purposes.

### Gait measurement

Patients performed a standardized barefoot treadmill gait analysis using the KneeKG™ system (Emovi, Montreal, Canada) one week before and three months after TKA. Patients selected their own treadmill velocity at each visit during a 3–5 minute adaptation phase without the KneeKG™ system. Once the patient was sufficiently at ease walking on the treadmill, the KneeKG™ harness was positioned and calibrated, and 45 seconds of the patient's gait were recorded.

### Kinematic conventions

Fully Cartesian coordinates were used to define non-orthogonal local coordinate systems (LCS) of the femur and tibia [10]. Knee kinematics computations followed International Society of Biomechanics conventions [11] using a previously described method [10]. Both joint angles (flexion–extension, adduction–abduction and internal–external rotation) and joint displacements (medial–lateral, anterior–posterior and superior–inferior) were computed.

## Computer-assisted surgery local coordinate systems

The femur's LCS during computer-assisted navigation was defined as follows (Fig 1). The proximal endpoint (**rPf**) was the hip joint centre, estimated using a functional method included in the Knee 3 CAS system (Brainlab, Munich, Germany). The distal endpoint (**rDf**) was the knee joint centre, estimated as the midpoint of the epicondyles projected onto the knee flexion–extension axis. The knee flexion–extension axis was estimated using the SARA method [12] and was used as the medial–lateral axis (**wf**) of the femur's LCS. This functional method was performed before and after definitive TKA, leading to a slightly different LCS at each time point. The anterior–posterior axis (**uf**) was defined as the normalised cross-product of the distal–proximal axis (**rPf-rDf**) by the medial–lateral axis (**wf**).

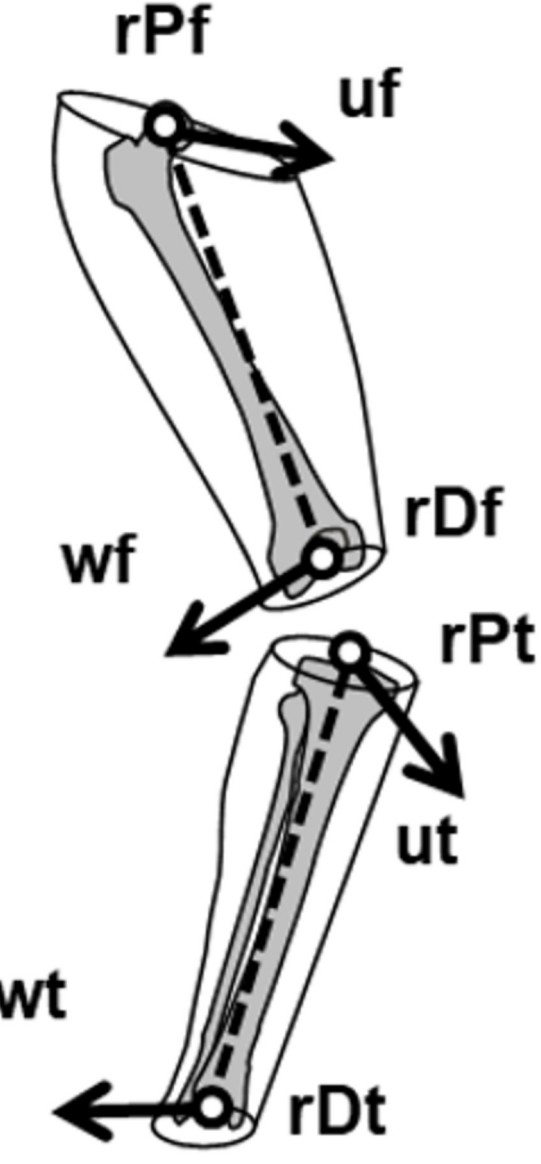

**Fig 1. Local coordinate systems of the femur and tibia.**

The tibia's LCS during computer-assisted navigation was defined as follows (Fig 1). The proximal endpoint (**rPt**) was the knee joint centre expressed as a point on the tibia at minimal flexion. The distal endpoint (**rDt**) was the midpoint of the malleoli. The anterior–posterior axis (**ut**) was aligned with the **uf** at minimal flexion. The medial–lateral axis (**wt**) was the normalised cross-product of the anterior–posterior axis (**ut**) and the distal–proximal axis (**rPt**-**rDt**).

## KneeKG™ local coordinate systems

The KneeKG™ system consists of a harness that reduces STAs and a calibration method that combines anatomical calibration, i.e. manual identification of anatomical points, and functional calibration, i.e. specific movements to identify axis and joint centres [5]. To avoid discrepancies in axis definition that would result in kinematic differences, the lower-limb geometry measured with the CAS system was introduced into the gait measurements using the calibration procedure described below.

First, a kinematic chain was defined based on the calibration of the CAS system, with a pivot at the knee and spherical joints at the hip and ankle. The pivot's axis was based on the flexion–extension axis (**wf**) estimated during CAS. The kinematic chain was introduced into the treadmill gait measurements using a two-level multi-body kinematics optimisation [13], where the variables to optimise were the fully Cartesian coordinates (**uf, rPf, rDf, wf, ut, rPt, rDt, wt**) and the positions of the reflective markers with respect to this kinematic chain. The outcome of the optimisation determined the optimal position of the kinematic chain, i.e. joint centres and flexion-extension axis, with respect to the thigh and shank clusters of the KneeKG™ system. These positions were used as the final calibration for the KneeKG™ system. In this way, the six degrees of freedom (DoF) kinematics were computed with the femur and tibia LCS matching those used during the CAS measurement.

## Data analysis

Each DoF measured using the CAS system was expressed as a function of the knee flexion–extension angle (coupling curves) [14]. Then, the CAS kinematics were expressed as a theoretical gait cycle by matching the CAS knee flexion measurement with the corresponding average knee flexion angle measured using the KneeKG™ system during gait (Fig 2). The matching was performed in four steps: 1) at each frame of the treadmill gait cycle, the knee flexion-extension angle was identified, 2) all the frames from the CAS measurements with this knee flexion-extension angle were identified, 3) the values of the different degrees of freedom of the knee at those frames were identified, 4) those values were reported as the CAS values at this instant of the gait cycle. In this way, the theoretical CAS kinematics during treadmill gait are determined and can be compared to the knee kinematics assessed with the KneeKG$^{TM}$. To increase the number of corresponding points between systems, the kinematic measurements from the CAS and the KneeKG™ were upsampled from 100 Hz to 300 Hz, and a distinction was made between the extension and flexion phases.

Each patient's adduction–abduction (AA) angle, internal–external rotation (IER) angle and anterior–posterior (AP) displacement [5], as measured during the treadmill gait test were averaged over the gait cycles and compared using a Bland–Altman analysis [15] to the corresponding CAS measurements averaged over the gait cycles. Next, bias and limits of agreement tests assessed how well their anatomical axes corresponded, while Spearman's correlation coefficient assessed the consistency of their kinematics pattern. Correlation coefficients were categorised as weak (0–0.30), moderate (0.31–0.50), good (0.51–0.70) and high ($> 0.70$) [16]. Each DoF's RoM was assessed as a reference for the limits of agreement. Finally, each system and patient's variability was assessed using the square root of the standard deviation (SD). A non-

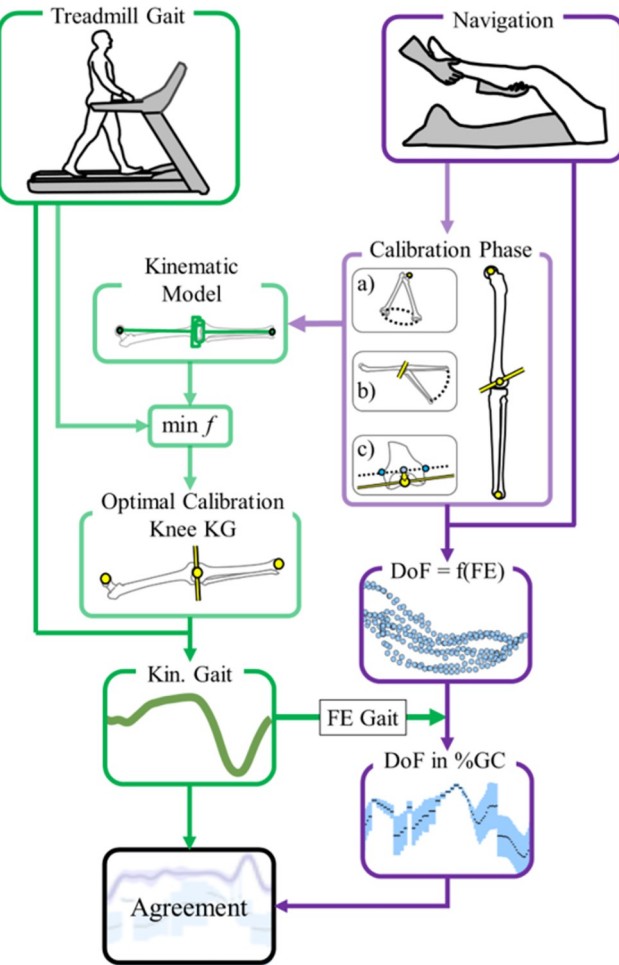

**Fig 2. Workflow of kinematic measurements and data analysis.** min $f$ represents the two-level optimisation methods used to fit the calibration of the CAS system on the gait measurement, DoF = f(FE) represents the degrees of freedom measured with the CAS system expressed in function of the flexion-extension of knee, FE Gait represents the matching of CAS knee flexion-extension with knee flexion-extension measured during gait to obtain the DoF in percentage of GC (DoF in %GC).

parametric Wilcoxon test was performed to assess differences in variability between systems. These analyses were performed over the whole gait cycle, for the single support phase and for the swing phase at two timepoints: before and after definitive TKA. Analyses at the two latter phases were selected to avoid any STAs due to foot contact in KneeKG™ measurements.

Calculations were made using the open-source Biomechanical ToolKit package [17], the 3D Kinematics and Inverse Dynamics toolbox [18], the Bland–Altman and Correlation Plot toolbox [19], and Matlab R2016b (MathWorks, USA). The workflow of measurements and data analysis is summarised in Fig 2.

## Results

This preliminary study included eight patients (mean ± SD: age, 70.4 ± 8.9 years; height, 161.9 ± 10 cm; weight, 78.6 ± 26.7 kg; 7 females), seven of whom underwent a TKA with a medial pivot design. Before surgery, all the patients walked more slowly on the treadmill than in their overground gait: 0.29 ± 0.19 m/s more slowly on average (Table 1).

**Table 1. Patient characteristics, the Hip Knee Ankle (HKA) angle was noted as positive for adducted knees (valgus) and negative for abducted knees (varus).** PRE = pre-surgery; P3M = three months post-surgery.

| | Sex | Age (years) | Weight (kg) | Height (cm) | BMI (kg/m²) | TKA side | HKA PRE (deg) | HKA P3M (deg) | Location OA | Femoral implant | Tibial implant | Patella implant | Overground Gait Speed (m/s) | Treadmill Gait Speed (m/s) |
|---|---|---|---|---|---|---|---|---|---|---|---|---|---|---|
| Patient 1 | F | 71.5 | 63.5 | 156.5 | 26 | R | -2 | 0 | FTI+FTE+FP | PFC | PFC | - | 0.94 | 0.89 |
| Patient 2 | F | 49.6 | 121 | 154.0 | 51 | L | -8 | -2 | FTI+FTE+FP | GMK SPHERE | GMK SPHERE | - | 1.11 | 0.56 |
| Patient 3 | F | 72.7 | 56 | 151.3 | 24 | L | -5 | 0 | FTI+FTE+FP | GMK SPHERE | GMK SPHERE | GMK | 0.94 | 0.56 |
| Patient 4 | F | 75.6 | 70 | 158.0 | 28 | R | -3 | 0 | FTI+FTE+FP | GMK SPHERE | GMK SPHERE | GMK | 1.03 | 0.97 |
| Patient 5 | M | 67.4 | 82 | 182.3 | 25 | L | -14 | -1 | FTI+FTE+FP | GMK SPHERE | GMK SPHERE | GMK | 1.36 | 1.11 |
| Patient 6 | M | 61.6 | 108 | 179.4 | 34 | R | -6 | - | FTI+FTE+FP | GMK SPHERE | GMK SPHERE | GMK | 1.25 | 0.69 |
| Patient 7 | F | 73.5 | 118 | 160.0 | 46 | R | -4 | 4 | FTI+FTE+FP | GMK SPHERE | GMK SPHERE | - | 0.64 | 0.39 |
| Patient 8 | F | 76.5 | 63 | 164.0 | 23 | L | -8 | 2 | FTI+FTE+FP | GMK SPHERE | GMK SPHERE | - | 0.83 | 0.58 |
| Median | - | 68.6 | 85.2 | 163.2 | 32 | - | -6 | 0 | - | - | - | - | 1.01 | 0.72 |
| IQR | - | 9.0 | 26.6 | 11.6 | 10.7 | - | 4 | 2 | - | - | - | - | 0.23 | 0.25 |

Fig 3 shows patient 1's knee kinematics curves as a typical example of the data used in the study. The kinematics of patients 2 to 8 are reported in S1 File. Table 2 reports on AA, IER and AP, bias, limits of agreements, RoM and SD, and Table 3 reports the number of patients per category of Spearman's correlation coefficient.

Before surgery, the variability in AA angle measured during treadmill gait was significantly lower than when measured during passive flexion–extension over the whole gait cycle (p = 0.008), in the single stance phase (p = 008) and in the swing phase (p = 0.008). On the contrary, after surgery, the variability in AP displacement measured during treadmill gait was

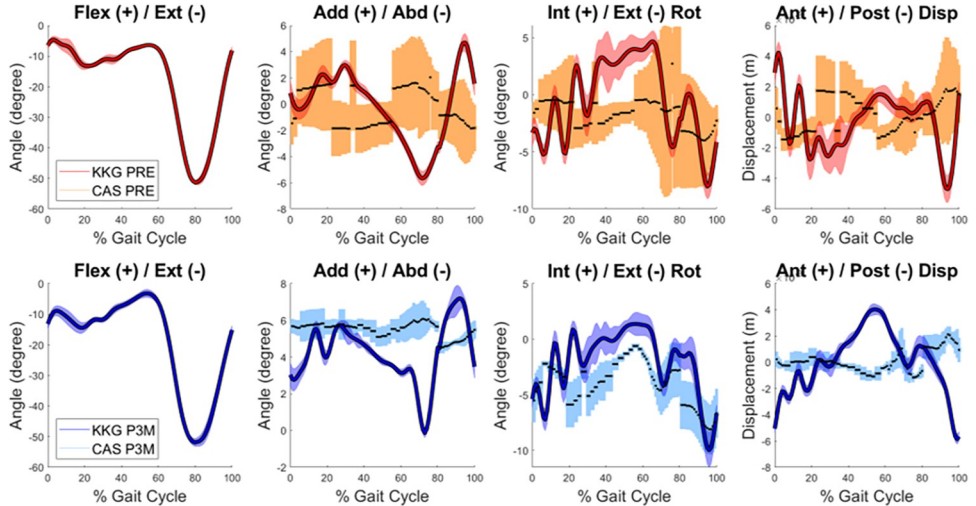

**Fig 3. Comparison of the degrees of freedom measured during computer-assisted navigation and treadmill gait assessement for patient 1, before (PRE) and after (P3M) surgery.**

**Table 2. Bias (b), limits of agreement (LoA), ranges of motion (RoM) and square root of the variance averaged over the gait cycle (SD) with KneeKG (KKG) and Computer-Assisted Surgery (CAS) for each degree of freedom and for the whole gait cycle, the single stance phase and the swing phase, both before and after TKA.** Results are reported in degrees as: Median (Inter-Quartile Range).

| | Pre-Surgery | | | Post-Surgery | | |
|---|---|---|---|---|---|---|
| | Gait Cycle | Single Stance | Swing Phase | Gait Cycle | Single Stance | Swing Phase |
| **Adduction–Abduction** | | | | | | |
| Bias | 0.1 (0.5) | -0.5 (1.5) | 0.6 (1.0) | -0.6 (1.2) | -1.2 (1.6) | -0.2 (1.0) |
| LoA | 4.4 (2.4) | 2.3 (1.5) | 5.3 (4.9) | 3.6 (1.1) | 1.7 (1.1) | 4.6 (2.0) |
| RoM KKG | 7.2 (4.8) | 3.3 (1.5) | 6.6 (5.4) | 6.6 (3.8) | 2.5 (1.9) | 6.4 (3.4) |
| RoM CAS | 3.0 (1.2) | 2.7 (1.7) | 2.9 (1.5) | 1.2 (0.7) | 0.8 (0.4) | 1.0 (0.7) |
| SD KKG | 0.5 (0.2) | 0.4 (0.2) | 0.5 (0.2) | 0.5 (0.2) | 0.3 (0.2) | 0.5 (0.2) |
| SD CAS | 1.8 (1.1) | 1.5 (1.3) | 1.8 (1.1) | 0.3 (0.2) | 0.3 (0.3) | 0.3 (0.2) |
| **Internal–External Rotation** | | | | | | |
| Bias | -1.1 (1.4) | -1.3 (1.7) | -1.2 (1.2) | -2.7 (1.6) | -3.1 (1.0) | -2.7 (2.5) |
| LoA | 3.5 (1.9) | 1.9 (2.6) | 4.2 (3.1) | 3.6 (1.3) | 3.1 (0.6) | 4.1 (2.0) |
| RoM KKG | 7.0 (3.4) | 3.4 (2.7) | 7.0 (3.2) | 8.0 (3.4) | 2.9 (2.3) | 7.6 (3.4) |
| RoM CAS | 5.9 (3.5) | 2.8 (1.7) | 5.7 (3.5) | 4.9 (2.9) | 3.7 (1.6) | 4.3 (1.9) |
| SD KKG | 1.7 (0.4) | 1.4 (0.3) | 1.9 (0.6) | 1.5 (0.5) | 1.5 (0.5) | 1.4 (0.6) |
| SD CAS | 2.0 (1.9) | 1.4 (1.6) | 3.2 (2.7) | 1.3 (0.8) | 2.0 (0.7) | 0.9 (0.7) |
| **Anterior–Posterior Displacement** | | | | | | |
| Bias | -0.7 (1.2) | -0.2 (2.8) | -0.5 (1.4) | -0.2 (0.6) | -0.2 (1.1) | -0.2 (0.9) |
| LoA | 3.3 (1.2) | 2.3 (1.1) | 3.2 (2.0) | 2.4 (2.1) | 2.2 (1.4) | 2.6 (2.2) |
| RoM KKG | 6.4 (1.3) | 3.6 (1.7) | 6.0 (1.5) | 4.6 (5.2) | 3.0 (2.5) | 3.9 (5.2) |
| RoM CAS | 3.3 (1.4) | 1.8 (1.8) | 2.9 (1.9) | 2.0 (1.5) | 1.3 (0.4) | 1.7 (1.1) |
| SD KKG | 0.8 (0.3) | 0.7 (0.4) | 0.8 (0.3) | 0.7 (0.2) | 0.7 (0.2) | 0.7 (0.3) |
| SD CAS | 1.0 (0.8) | 0.7 (0.5) | 1.2 (1.1) | 0.5 (0.1) | 0.6 (0.2) | 0.3 (0.2) |

significantly higher than during passive flexion–extension, but only when considering the whole gait cycle ($p = 0.023$). There were no other significant differences regarding variability.

**Table 3. Number of patients in each category of $R^2$ for each degrees of freedom and for the whole gait cycle (GC), the single stance phase (SS) and the swing phase (SW) before and after TKA.**

| | Pre-surgery | | | Post-surgery | | |
|---|---|---|---|---|---|---|
| | GC | SS | SW | GC | SS | SW |
| **Adduction–Abduction** | | | | | | |
| n weak | 7 | 6 | 3 | 7 | 3 | 4 |
| n moderate | 0 | 1 | 2 | 1 | 3 | 2 |
| n good | 1 | 0 | 3 | 0 | 1 | 0 |
| n high | 0 | 1 | 0 | 0 | 1 | 2 |
| **Internal–External Rotation** | | | | | | |
| n weak | 5 | 6 | 3 | 6 | 7 | 3 |
| n moderate | 2 | 2 | 2 | 2 | 0 | 2 |
| n good | 1 | 0 | 3 | 0 | 0 | 3 |
| n high | 0 | 0 | 0 | 0 | 1 | 0 |
| **Anterior–Posterior Surgery** | | | | | | |
| n weak | 7 | 8 | 4 | 7 | 5 | 5 |
| n moderate | 1 | 0 | 1 | 1 | 1 | 2 |
| n good | 0 | 0 | 3 | 0 | 1 | 1 |
| n high | 0 | 0 | 0 | 0 | 1 | 0 |

## Discussion

This preliminary study investigated associations between passive knee kinematics measured using CAS before and after a definitive TKA and active knee kinematics during a treadmill gait assessment before and three months after surgery. The low biases calculated showed how well the anatomical axes identified with each system corresponded with each other. Indeed, before and after surgery biases were close to one degree and to one millimetre for the AA angle and AP displacement, respectively, and below three degrees for the IER angle. This is supported by a recent study showing a strong positive correlation (R = 0.66) between AA angle measurements at minimal knee flexion during computer-assisted navigation and during gait analysis [20]. These results indicate that when a knee is more adducted and internally rotated during computer-assisted surgery, the same general alignment can be observed during gait assessment and vice versa. The LoA of the AA and IER angles seemed high since they were of the same order of magnitude as their RoM during walking. Nevertheless, a previous study comparing standard marker-based methods to gold-standard kinematics measurement methods found slightly higher LoA for AA angles (5.2˚), IER angles (4.1˚) and AP displacements (10.1 mm) during gait assessment [21]. It seems that the calibration of the KneeKG™ system employing a kinematic chain defined using computer-assisted navigation performed well compared to standard marker-based methods. Still, the mean AA angle three months post-surgery seemed largely overestimated during the treadmill measurement when compared to the outcomes of CAS.

The majority of individual correlations between the CAS system data and treadmill gait data were weak over the whole gait cycle. Correlations were better at the phase level and better during the swing phase than during the single support phase. Indeed, half of the correlations for the swing phase were moderate to high, whereas only a third of the correlations in the single support phase were in these categories. This indicates that although the optimised calibration and static alignment seemed right, kinematics patterns between the measurements were not always consistent. Likewise, Roda *et al.* found no statistically significant associations between the preoperative varus thrust (range of the AA angle in early stance), as measured using standard marker-based methods, and the peak and range of the AA angle measured intraoperatively [22].

In the present study, the variability of the AA angle measured during pre-surgery treadmill gait was lower than those in the CAS measurements. This could suggest that muscle control under weight-bearing conditions can help to stabilise an arthritic knee joint. Interestingly, larger discontinuities and SDs in the AA angle and AP displacement were observed in the computer-assisted navigation data before TKA than after it. This may indicate that end-stage knee OA can lead to potential knee instability, as underlined by the greater variability in those two DoF. This kinematics variability in joint kinematics also reflects the variability in the external loads magnitude and direction applied manually by the surgeons during passive flexion-extension. Post-TKA evaluations showed less variability and a more consistent pattern, which could indicate increased passive stability after TKA.

The CAS and KneeKG™ systems present several sources of differences. First, regarding anatomical and biomechanical differences, during CAS, knee kinematics are assessed after arthrotomy, passively and without loading, whereas gait kinematics are assessed with a closed knee, actively and with loading and impacts. One recent study suggested that there were no differences between active and passive knee flexion–extension angles after arthrotomy but without loading [7]. Moreover, good correlations were found between CAS measurements and impact-free weight-bearing knee kinematics assessed using monoplane fluoroscopy six months after surgery [8, 9]. These results suggest that the differences observed in the present

study could come from the weight-bearing conditions pre-surgery, from the impact conditions post-surgery and from measurement errors. Regarding the second source of differences—i.e. measurement methods and systems—both systems (CAS and KneeKG™) use single beam optoelectronic cameras (Northern Digital, Ontario, Canada) and a combination of anatomical and functional calibration. An optimisation procedure was used to minimise the calibration differences between the two systems. Thus, the main technical difference seems to come from how the clusters of reflective markers were attached to the lower limb. Indeed, during CAS, each cluster is fixed to two intracortical pins, whereas during the gait assessment, the clusters are fixed to two harnesses attached on the skin. Although the femur harness tends to reduce STAs [23–25] in quasi-static and non-weight-bearing conditions, some errors may remain, leading to differences in gait kinematics assessment. A previous study of obese subjects evaluated root-mean-square errors of between 1.2˚ and 3.0˚ for the three angles during quasi-static squats and of between 4.4 mm and 8.9 mm for the three displacements, with values from 1.4˚ to 8.0˚ and 5.2 mm to 9.1 mm for non-obese subjects [25]. Due to these multiple sources of differences, this preliminary study could not prove that knee kinematics during passive movement and treadmill gait were different. Answering this question would require comparing the gold-standard CAS measurement with the gold-standard gait measurement (i.e. biplane fluoroscopy).

From a methodological standpoint, apart from the optimised calibration of the KneeKG™ system, using the mean knee flexion–extension angle to compare passive flexion–extension and gait kinematics allowed us to make a simple comparison between movements using measurements of their different amplitudes, durations and velocities. This was similar to the method used by Deroche *et al.* [20] but extended over the whole gait cycle. Our method was specifically designed to assess the potential of using CAS measurements to predict gait kinematics, whereas other studies had used RoM and peaks [22] or coupling curves between flexions and the other DoFs [8, 9]. The fact that the bias and LoA found in the present study were consistent with the reported errors due to STAs also suggests that the passive kinematics measured during CAS may potentially serve as constraints [21, 26] to compensate for STA.

This preliminary study had some limitations. First, the number of subjects was rather low: a larger database would provide additional information and enable a better understanding of the differences observed. Second, the treadmill was a significant limitation because patients were not used to a treadmill gait and, due to pain and age, the adaptation phase before performing the measurements had to remain short (3–5 minutes). Consequently, most patients showed significant differences between their treadmill and overground gait, as illustrated by the differences in speed (median 1.01 m/s overground vs median 0.72 m/s on the treadmill). Third, intraoperative recordings were made after the arthrotomy and before closing the opening. Indeed, the calibration of the distal femur and the proximal tibia required an open joint. The knee's central pivot (anterior and posterior cruciate ligaments) was left untouched for the acquisition of passive RoM under OA conditions, thus allowing for data as close as possible to the native situation. Nevertheless, arthrotomy could potentially bias the DoF compared to a closed joint. In future studies, the incision could be sutured before acquiring movements, but this would require longer surgery time.

## Conclusions

The optimised calibration used in this preliminary study led to closely corresponding anatomical axes of the knee when comparing measurements from computer-assisted surgery and the KneeKG™ system, i.e. knees found to be more adducted and internally rotated during computer-assisted surgery presented with the same general alignment during gait assessments.

However, it seems that the consistency between the systems' kinematics patterns was low. The difference in variability between the active and passive measurements taken before surgery suggests that muscle control may stabilise the knee during a gait assessment and that the two approaches are complementary. The multiple sources of differences meant that we were unable to understand whether they came from true anatomical and biomechanical differences or from measurement discrepancies. A comparison of computer-assisted surgery measurement and gait measured using a gold-standard motion-capture system (e.g. biplane fluoroscopy) would be necessary to answer this question.

## Supporting information

**S1 File. Comparison of the degrees of freedom measured during computer-assisted navigation and treadmill gait assessement for patient 2 to 8, before (PRE) and after (P3M) surgery.**
(PDF)

## Author Contributions

**Conceptualization:** Xavier Gasparutto, Alice Bonnefoy-Mazure, Raphaël Dumas.

**Data curation:** Xavier Gasparutto, Alice Bonnefoy-Mazure, Michael Attias, Hermès Miozzari.

**Formal analysis:** Xavier Gasparutto, Alice Bonnefoy-Mazure, Raphaël Dumas, Stéphane Armand.

**Funding acquisition:** Hermès Miozzari.

**Investigation:** Xavier Gasparutto, Hermès Miozzari.

**Methodology:** Xavier Gasparutto, Alice Bonnefoy-Mazure, Michael Attias, Raphaël Dumas, Stéphane Armand.

**Project administration:** Michael Attias, Hermès Miozzari.

**Software:** Raphaël Dumas.

**Supervision:** Stéphane Armand, Hermès Miozzari.

**Writing – original draft:** Xavier Gasparutto, Alice Bonnefoy-Mazure.

**Writing – review & editing:** Xavier Gasparutto, Alice Bonnefoy-Mazure, Michael Attias, Raphaël Dumas, Stéphane Armand, Hermès Miozzari.

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
