## [Decision Letter · Decision Letter 0]

14 Jul 2022

PONE-D-21-38814Comparison between passive knee kinematics during surgery and active knee kinematics during walking: a preliminary studyPLOS ONE

Dear Dr. Gasparutto,

Thank you for submitting your manuscript to PLOS ONE. After careful consideration, we feel that it has merit but does not fully meet PLOS ONE’s publication criteria as it currently stands. Therefore, we invite you to submit a revised version of the manuscript that addresses the points raised during the review process. Two reviewers provided input for the assessment of your manuscript. Please carefully address their comments. The reviewers particularly agree that the sample size is low, and that it cannot serve as a basis for meaningful conclusions. Therefore, a revised version should focus more on explanation of methods and tone down conclusions from the study (which might better be called a pilot), besides leading to formulation of protocols and hypotheses that remain to be tested. Furthermore, I could not find any information on pre-registration of this study and its protocol. Please add that information. Please submit your revised manuscript by Aug 28 2022 11:59PM. If you will need more time than this to complete your revisions, please reply to this message or contact the journal office at plosone@plos.org. Please include the following items when submitting your revised manuscript:A rebuttal letter that responds to each point raised by the academic editor and reviewer(s). You should upload this letter as a separate file labeled 'Response to Reviewers'.A marked-up copy of your manuscript that highlights changes made to the original version. You should upload this as a separate file labeled 'Revised Manuscript with Track Changes'.An unmarked version of your revised paper without tracked changes. You should upload this as a separate file labeled 'Manuscript'.

We look forward to receiving your revised manuscript.

Kind regards,

Heike Vallery

Academic Editor

PLOS ONE

Journal Requirements:

“This work was supported by Geneva University Hospitals’ Department of Orthopaedic Surgery and Trauma Care. The funding source played no role in the study’s design.”

3. Thank you for stating the following in the Competing Interests/Financial Disclosure* (delete as necessary) section:

“I have read the journal's policy and the authors of this manuscript have the following competing interests: Hermes Miozzari is an associate editor for EFORT Open Reviews and a board member of Swiss Orthopaedics. Stéphane Armand is a member of the editorial board of EFORT Open Reviews.

There are no competing interests associated with this research.”

We note that one or more of the authors are employed by a commercial company: EFORT Open Reviews and Swiss Orthopaedics

“This work was supported by Geneva University Hospitals’ Department of Orthopaedic Surgery and Trauma Care. The funding source played no role in the study’s design.”

“This work was supported by Geneva University Hospitals’ Department of Orthopaedic Surgery and Trauma Care. The funding source played no role in the study’s design.”

Reviewers' comments:

Reviewer's Responses to Questions

**Comments to the Author**

1. Is the manuscript technically sound, and do the data support the conclusions?

Reviewer #1: Partly

Reviewer #2: Yes

2. Has the statistical analysis been performed appropriately and rigorously? 

Reviewer #1: No

Reviewer #2: I Don't Know

3. Have the authors made all data underlying the findings in their manuscript fully available?

Reviewer #1: Yes

Reviewer #2: Yes

4. Is the manuscript presented in an intelligible fashion and written in standard English?

Reviewer #1: Yes

Reviewer #2: Yes

5. Review Comments to the Author

Reviewer #1: Thank you for reviewing this interesting study. Although the reviewer understands this is a preliminary study, the limited number of patients (n=8) cannot add any information about the comparison between passive knee kinematics during surgery and active knee kinematics during walking.

Reviewer #2: This study aims to develop a relationship between passive knee kinematics obtained during surgery and active kinematics during gait following surgery. This is an interesting idea, which immediately brings several concerns to mind, but it would be a valuable contribution for surgeons. The idea is an excellent one.

The article is also well written and with well articulated objectives.

One paragraph still has me struggling to understand. Lines 159-165, where the optimization of the coordinates used in the CAS system were determined in the KneeKG system, in order to compare the two kinematic outputs. I don’t quite follow this, and I am wondering how a comparison can be made between passive knee flexion extension and gait.

Line 170 – reference should be to figure 2, not figure 1

Line 172 - How and why was the upsampling done?

Line 174 – “mean AA angle, IE angle and AP displacement” – why these variables specifically? Does “mean” indicate that a mean angle was taken from the entire gait cycle, or from a specific knee flexion extension angle?

Line 190 – I think it would be helpful to understand if more detail were added to the caption for Figure 2, explaining in particular what data is shown in DoF = f(FE), how this is impacted by FE Gait, and again DoF in %GC.

Table 1 title – please define hka – hip knee ankle

I make the above suggestions since the value in this manuscript to readers is in understanding the methods the authors have used to relate the two distinctly derived knee kinematic measures. No other real conclusions can be drawn from this manuscript due to the low sample size, but this is appropriate for a preliminary study such as this one.

6. PLOS authors have the option to publish the peer review history of their article (what does this mean?). If published, this will include your full peer review and any attached files.

Reviewer #1: No

Reviewer #2: No

---

## [Author Response · Author response to Decision Letter 0]

25 Aug 2022

The response to reviewer was attached as a separate file.

---

## [Editor Report · Decision Letter 1]

31 Aug 2022

PONE-D-21-38814R1Comparison between passive knee kinematics during surgery and active knee kinematics during walking: a preliminary studyPLOS ONE

Dear Dr. Gasparutto,

Thank you for submitting your manuscript to PLOS ONE. **I noticed that the data is not yet available for review, so I have not sent this manuscript out to reviewers yet. Please resubmit your files including a link to the data and code, such that the reviewers have access. Of course, this may at this stage be a private link. ** Please submit your revised manuscript by Oct 15 2022 11:59PM. If you will need more time than this to complete your revisions, please reply to this message or contact the journal office at plosone@plos.org. Kind regards,

Heike Vallery

Academic Editor

PLOS ONE
---

## [Author Response · Author response to Decision Letter 1]

30 Sep 2022

The response to the reviewers comments are detailed in the document "NK - Response to reviewers_Final.DOCX".

Best regards,

The authors

---

## [Decision Letter · Decision Letter 2]

14 Nov 2022

PONE-D-21-38814R2Comparison between passive knee kinematics during surgery and active knee kinematics during walking: a preliminary studyPLOS ONE

Dear Dr. Gasparutto,

Thank you for submitting your manuscript to PLOS ONE. After careful consideration, we feel that it has merit but does not fully meet PLOS ONE’s publication criteria as it currently stands. Therefore, we invite you to submit a revised version of the manuscript that addresses the points raised during the review process. Reviewers agree that this is an interesting study. One of the reviewers still has several comments that need addressing in a revision.The main concern of this manuscript remains the small study size and limited possibility to draw conclusions. So, conclusions of the manuscript should be toned down more, towards formulating hypotheses that could serve as the basis for a bigger study.

We look forward to receiving your revised manuscript.

Kind regards,

Heike Vallery

Academic Editor

PLOS ONE

Journal Requirements:

Reviewers' comments:

Reviewer's Responses to Questions

**Comments to the Author**

1. If the authors have adequately addressed your comments raised in a previous round of review and you feel that this manuscript is now acceptable for publication, you may indicate that here to bypass the “Comments to the Author” section, enter your conflict of interest statement in the “Confidential to Editor” section, and submit your "Accept" recommendation.

Reviewer #2: All comments have been addressed

Reviewer #3: (No Response)

2. Is the manuscript technically sound, and do the data support the conclusions?

Reviewer #2: Yes

Reviewer #3: Partly

3. Has the statistical analysis been performed appropriately and rigorously? 

Reviewer #2: Yes

Reviewer #3: Yes

4. Have the authors made all data underlying the findings in their manuscript fully available?

Reviewer #2: Yes

Reviewer #3: Yes

5. Is the manuscript presented in an intelligible fashion and written in standard English?

Reviewer #2: Yes

Reviewer #3: Yes

6. Review Comments to the Author

Reviewer #2: (No Response)

Reviewer #3: This study sought to present a detailed comparison between passive knee kinematics (i.e., passive flexion-extension) acquired during total knee arthroplasty (TKA) by means of computer assisted-surgery system and active kinematics measured during walking with a marker-based optoelectronic system (KneeKG). The analysis was performed on 8 patients, who underwent TKA. Active kinematics was acquired before surgery and 3 months after, whereas passive kinematics was measured before and after TKA implantation. Anatomical axes were homogenised by using 2-levels optimization based on multi-body modelling, so as to limit any bias between active and passive kinematics. The authors compared knee joint angles acquired before and after TKA. The authors underlined very weak correlations between active and passive kinematics and they were not able to identify the source of errors.

General comment

The hypothesis at the basis of this paper is clearly reported as far as the main objective. Both the experimental phase and data analysis are written with a good level of details; the analysis in particular was performed well, and the synthesis of the obtained results is appreciable. Although the methodology in itself is not that innovative, the comparison provides useful hints and novel perspective for the application field.

The structure of the article seems to be precise (Abstract, Introduction, Methodology [with subheadings], Results, Discussion and Conclusions).

Experimental phase and data analysis seem to be clearly reported and are coherent with the work objectives. Several concerns have been reported to the authors.

The use of the English language seems to be correct.

Specific Comments

Title

Ok.

Abstract

In general, this section is quite ok.

Line 35-37: Please give few more hints concerning the “homogenization” technique.

Keywords

I would not use the name of a commercial system (KneeKG) as a keyword for a scientific work.

Introduction

Ok, well identified the main hypotheses and defined the main goal.

Materials and Methods

Ok, the experimental phase and data analysis was reported with a good level of detail.

Since the study was approved by an Ethical Committee, could you please give some information about the identification of the correct sample size? Did you perform any power analysis? How this led you to the possibility to generalize your results?

The passive flexion-extension was performed while loading the limb from the foot? Did you think loads could influence your results?

How did you identify the “gait cycle” over the passive

Results

Ok.

Discussion

Ok. Thank you for underling the main limitations of your work.

Conclusions

Ok. My only concern is related to the possibility to generalize your results with such a small sample size.

References

The references to previous works seem to be precise, wide and up-to-date.

Figures

Ok.

Tables

Ok.

7. PLOS authors have the option to publish the peer review history of their article (what does this mean?). If published, this will include your full peer review and any attached files.

Reviewer #2: No

Reviewer #3: **Yes: **Nicola Francesco Lopomo

---

## [Decision Letter · Decision Letter 3]

17 Feb 2023

Comparison between passive knee kinematics during surgery and active knee kinematics during walking: a preliminary study

PONE-D-21-38814R3

Dear Dr. Gasparutto,

We’re pleased to inform you that your manuscript has been judged scientifically suitable for publication and will be formally accepted for publication once it meets all outstanding technical requirements.

Kind regards,

Emiliano Cè

Academic Editor

PLOS ONE

Additional Editor Comments (optional):

Reviewers' comments:

Reviewer's Responses to Questions

**Comments to the Author**

1. If the authors have adequately addressed your comments raised in a previous round of review and you feel that this manuscript is now acceptable for publication, you may indicate that here to bypass the “Comments to the Author” section, enter your conflict of interest statement in the “Confidential to Editor” section, and submit your "Accept" recommendation.

Reviewer #2: (No Response)

2. Is the manuscript technically sound, and do the data support the conclusions?

Reviewer #2: Partly

3. Has the statistical analysis been performed appropriately and rigorously? 

Reviewer #2: Yes

4. Have the authors made all data underlying the findings in their manuscript fully available?

Reviewer #2: Yes

5. Is the manuscript presented in an intelligible fashion and written in standard English?

Reviewer #2: Yes

6. Review Comments to the Author

Reviewer #2: (No Response)

7. PLOS authors have the option to publish the peer review history of their article (what does this mean?). If published, this will include your full peer review and any attached files.

Reviewer #2: No

---

## [Editor Report · Acceptance letter]

24 Feb 2023

PONE-D-21-38814R3 

COMPARISON BETWEEN PASSIVE KNEE KINEMATICS DURING SURGERY AND ACTIVE KNEE KINEMATICS DURING WALKING: A PRELIMINARY STUDY 

Dear Dr. Gasparutto:

I'm pleased to inform you that your manuscript has been deemed suitable for publication in PLOS ONE. Congratulations! Your manuscript is now with our production department. 

Kind regards, 

on behalf of

Professor Emiliano Cè 

Academic Editor

PLOS ONE